# Development of an Immunocapture-Based Polymeric Optical Fiber Sensor for Bacterial Detection in Water

**DOI:** 10.3390/polym16060861

**Published:** 2024-03-21

**Authors:** Rafaela Nascimento Lopes, Paulo Henrique Silva Pinto, Juan David Lopez Vargas, Alex Dante, Andrew Macrae, Regina Célia Barros Allil, Marcelo Martins Werneck

**Affiliations:** 1Electrical Engineering Program, COPPE, Universidade Federal do Rio de Janeiro, Rio de Janeiro 21941-598, Brazil; raphaelanlopes@gmail.com (R.N.L.); paulo.hsp11@gmail.com (P.H.S.P.); alexdante@coppe.ufrj.br (A.D.); reginaallil@coppe.ufrj.br (R.C.B.A.); 2Programa de Pós-Graduação em Biotecnologia Vegetal e Bioprocessos, CCS, Universidade Federal do Rio de Janeiro, Rio de Janeiro 21941-598, Brazil; amacrae@micro.ufrj.br; 3Nanotechnology Engineering Program, COPPE, Universidade Federal do Rio de Janeiro, Rio de Janeiro 21941-598, Brazil; chivirri@utp.edu.co

**Keywords:** plastic optical fiber sensor, immunosensor, *Escherichia coli*, biosensor

## Abstract

Conventional methods for pathogen detection in water rely on time-consuming enrichment steps followed by biochemical identification strategies, which require assay times ranging from 24 hours to a week. However, in recent years, significant efforts have been made to develop biosensing technologies enabling rapid and close-to-real-time detection of waterborne pathogens. In previous studies, we developed a plastic optical fiber (POF) immunosensor using an optoelectronic configuration consisting of a U-Shape probe connected to an LED and a photodetector. Bacterial detection was evaluated with the immunosensor immersed in a bacterial suspension in water with a known concentration. Here, we report on the sensitivity of a new optoelectronic configuration consisting of two POF U-shaped probes, one as the reference and the other as the immunosensor, for the detection of *Escherichia coli*. In addition, another methos of detection was tested where the sensors were calibrated in the air, before being immersed in a bacterial suspension and then read in the air. This modification improved sensor sensitivity and resulted in a faster detection time. After the immunocapture, the sensors were DAPI-stained and submitted to confocal microscopy. The histograms obtained confirmed that the responses of the immunosensors were due to the bacteria. This new sensor detected the presence of *E. coli* at 10^4^ CFU/mL in less than 20 min. Currently, sub-20 min is faster than previous studies using fiber-optic based biosensors. We report on an inexpensive and faster detection technology when compared with conventional methods.

## 1. Introduction

Waterborne bacterial, viral, and parasitic pathogens are a global health problem. Lack of access to safe drinking water combined with poor hygiene and sanitation facilities affects more than half of the population in developing countries [1]. About one billion people depend on contaminated water sources, resulting in about 2.2 million deaths annually, mainly caused by diarrheal diseases, which the World Health Organization (WHO) estimates to account for about 4% of the global burden of disease [2].

The detection of pathogens in water is complicated by several obstacles: they are usually present in very low concentrations in the environment and the samples contain numerous inhibitors of enzymatic reactions as well as interfering organisms and particles [2]. Conventional methods for pathogen detection rely on time-consuming enrichment steps followed by biochemical identification strategies that require assay times off 24 hours to one week [3]. In recent years, however, considerable efforts have been made to develop biosensing technologies that enable the rapid and near real-time detection of pathogens in water.

A biosensor is a self-contained analytical device consisting of a biological recognition element and a transducer. The analyte, e.g., the bacteria, binds to the biological element, which in turn generates an electronic response in the transducer that can be measured [2]. Optical biosensors using a variety of optical sensing modalities have been promoted as a promising alternative transducer platform for pathogen analysis.

Biosensors can be classified according to the principle of operation under which the transducer works or the type of the bioreceptor. The principle of operation ranges from (a) optical to (b) electrochemical, (c) mass-sensitive, or (d) thermometric. Bioreceptor elements can vary from (a) proteins that catalyze specific chemical reactions; to (b) antibodies and antigens based on the antibody–antigen interaction in which a specific antibody binds to a specific antigen; (c) nucleic acids; (d) biomimetic receptors in which recognition is achieved through the use of imprinted polymers that mimic the bioreceptor; and (f) whole cells or a specific cellular component [4]. In general, biosensors can detect even minor changes in analytes, enabling sensitive and specific measurements [3].

An example of an electrochemical biosensor can be found in the work of Sobhan et al. (2022) [5]. In their research, they developed an activated biochar-based immunosensor for the detection of *Escherichia coli* (*E. coli*) cells in a pure culture. They immobilized *E. coli* antibodies on the surface of the electrodes and measured the impedance of the immunosensor using an impedance analyzer. They were able to detect *E. coli* strain O157:H7 at 10^4^ CFU/mL.

Biosensors based on antibody–antigen interactions are also known as immunosensors where the biorecognition element is an antibody. These types of biosensors are highly selective and can recognize a specific antigen or bacterium among many other species. To function properly, the antibodies are immobilized on the sensor, or on the surface of an optical fiber, as we will describe later in this article.

In recent years, optical fibers have been successfully used as immunosensor platforms because of their fast response, specificity, sensitivity, and low cost. In addition, they are suitable for near real-time monitoring and on-site detection, as shown in Wandermur et al. (2013) [6].

One of the preferred physical parameters of a fiber optic sensor is the refractive index (RI), which changes depending on the external environment and can be used as a sensing basis, but many studies present different parameters used for detection, as described by Wang and Wolbeis (2020) [7], who provide an overview of these sensors.

For example, Razo-Medina et al. (2018) [8] have described a biosensor for cholesterol based on the thin film of a cholesterol enzyme encapsulated in a sol-gel film applied to the end of a plastic optical fiber.

In a fiber-optic biosensor, the immobilization of the enzymes on the fiber surface is normally achieved through chemical modification. Li et al. (2021) [9] immobilized an enzyme on an optical fiber sensor for the detection of glucose.

There are many examples that show different techniques applying optical fiber sensors; for a more detailed description of these methodologies, see Leitão et al. (2022) [10]. The paper describes cost-effective fiber configurations such as end-face, reflected, uncladded, D-shape, U-shape, tips, tapered, amongst others.

Another way to apply an optical fiber as a sensor is based on the surface plasmon resonance (SPR) phenomena. Traditional SPR systems use a thin metal film deposited on the surface of a prism. A monochromator measures the intensity and wavelength of the reflected light at the thin metal film. The light is more attenuated (absorbed by the electron resonance) at a specific angle and a specific wavelength that depends on the metal film used. Many factors can change the wavelength, such as refractive index of a liquid in contact with the thin film or the presence of molecules attached to the film [11].

Since the late 1970’s, SPR was believed to be useful to characterize thin films and to monitor chemical process occurring over the thin film. Nylander et al. (1982) [12] were the first to apply SPR for gas detection and biosensing.

Later, Mitsushio et al. (2006) [13] deposited 45 nm thin films of Au, Ag, Cu, and Al on the surface of an optical fiber and thus developed an SPR-based optical fiber sensor for the first time. After this first application, many articles have been published with a variety of sensing applications, such as Arcas et al. (2018) [14], who detected *E. coli* with a U-shape plastic optical fiber covered with an Au nanofilm, Arcas et al. (2021) [15], who detected *Taenia solium*, the pork tapeworm that causes intestinal infection after eating contaminated pork, Cennano et al. (2021) [16], detected SARS-CoV-2 with an SPR-based optical fiber sensor, and Alberti et al. (2022) [17], who detected uranium in water.

Plastic optical fibers (POF) were first developed by DuPont in 1963. They were initially used for lighting over short distance. Today, we use conventional POFs made of polymethyl methacrylate (PMMA), which were also developed by DuPont in 1968 [18].

In comparison with silica fibers, POFs have not evolved much in terms of reducing their transmission losses over distances. Owing to these limitations, POFs are restricted to short distances for telecommunication applications, up to100 m. That noted while POFs have limited use for telecommunications, they are interesting for sensor development.

The reason for this is that POFs offer numerous advantages over silica fibers, including a larger diameter that facilitates handling, good light coupling, and the use of low-cost peripheral components that can be connected to readily available transmission components such as LEDs and photodetectors at a low cost and using simple tools.

With these unique properties, POFs have been applied in various sensor applications, such as chemical and biological sensing as well as those of strain, temperature, and displacement [18].

Plastic optical fiber sensors have been the focus of research in our lab. One of our first studies was conducted by Beres et al. (2011) [19] involving the detection of *E. coli* in water using a tapered POF sensor. Subsequently, Wandermur et al. (2014) [6] developed a U-Shaped POF sensor in an electronic platform for the rapid detection of bacteria. Following these studies, Rodrigues et al. (2017) [20] investigated the sensitivity of different forms of a U-Shaped POF sensor and searched for a better efficiency at low bacterial numbers, while Lopes et al. (2018) [21] used a specific U-Shape sensor format for the detection of the sulfate-reducing bacteria, *Desulfovibrio alaskensis*, which is found in crude oil and responsible for the production of hydrogen sulfite (H_2_S), which reacts with water and produces sulfuric acid (H_2_SO_4_), souring the oil and corroding pipelines.

Ashraf et al. (2022) [22] describes the use of U-Shaped POF s to detect iron, and, in Ashraf et al. (2023) [23], to detect phosphate in water.

Of note is the work of Johari et al. (2022) [24], who used a tapered U-Shaped POF sensor coated with ZnO nanorods to measure relative humidity, and the work of Hadi and Khurshid (2022) [25], who used a U-Shaped POF immunosensor for the detection of SARS-CoV-2.

The above-mentioned U-Shaped POF sensors use one of the most common operating principles of POF immunosensors, namely, the change in guided light output at the fiber end in response to the pathogens captured by the immobilized antibodies on the fiber surface. In previous studies, we have developed a POF immunosensor using an optoelectronic configuration consisting of a U-shaped probe connected to an 880 nm LED and a photodetector [21,26].

In this article, we report a new development in which we used a new reading method and an improved electronic system consisting of two POF U-shaped probes, one as a reference and the other as an immunosensor. We evaluate the sensitivity of this novel optoelectronic configuration for the detection of *E. coli*.

## 2. Sensing Theory: Physical Principle of a U-Shaped POF Sensor

This section explains the mechanism by which the developed immunosensor works and the theories associated with this sensing method. The sensor is based on the change in the refractive index (RI) within the evanescent wave created near the fiber surface by the bacteria captured through the antigen–antibody effect.

Figure 1 shows a schematic representation of the sensing principle using a U-Shaped POF sensor immersed in a liquid containing several species of bacteria. A near-infrared LED is used to couple light into an uncladded poly (methyl methacrylate) (PMMA) fiber, and the light is transmitted through the fiber. A microcontroller controls the optical output power of the LED, which is coupled into the sensor. The light is guided through the fiber to the photodetector, which converts the light into a photocurrent, which is then converted into voltage levels that are properly conditioned and detected by the microcontroller. The fiber surface is functionalized with a specific antibody that performs the immunocapture process specifically for *E. coli* bacteria. As bacteria are attracted and captured on the fiber surface, the RI of the interface between the fiber core and the outer medium changes, which in turn changes the guiding conditions of the fiber. As a result, the optical output power at the fiber end varies depending on the number of bacteria captured by the antibody.

## 3. Materials and Methods

### 3.1. Manufacture of U-Shape Sensors

The POF used to manufacture the proposed immunosensor has a core diameter of 980 µm with a 10 µm cladding, made of poly (methyl methacrylate) (PMMA) (Mitsubishi model CK40 Eska^®^, Tempe, AZ, USA). In previous studies, we observed that the fiber cladding prevented the good functionalization and immobilization of the antibodies on its surface because it is made of PMMA and other polymers that renders the RI below that of the core [21]. Therefore, we removed the 10 μm thick cladding by the following procedure: the curve of the sensor was placed inside the folding of an optical cleaning tissue, applied 50 μL of acetone and the fiber curve is gently hand-rubbed. After that, the sensor was rinsed in distilled water to neutralize the effect of corrosion [20].

For sensor fabrication, the fiber was cleaved into 10 cm long sections and both end surfaces were polished by a 1500-grit sandpaper for a better light coupling. Following that, the fiber sections were rinsed with deionized water and blow-dried with nitrogen. Then, the POFs were bent around a 10 mm width 3-D printed mold to produce the U-Shape probes, and heated in an oven at 70 °C. The sensors were tested under different RIs, to check for reproducibility and to calibrate their sensitivity, and further functionalized with antibodies. Twenty sensors were fabricated to produce 10 reference sensors and 10 immunosensors. Five pairs were used to detect bacteria in a 10^8^ CFU/mL (Colony Forming Unities per milliliter) suspension and the other five pairs were used to detect bacteria within 10^4^ CFU/mL suspension.

### 3.2. Sensor Surface Functionalization with Polyethyleneimine (PEI) and Immobilization of Antibodies

The following protocol modified from [27] was adopted to functionalize the probes: The sensors were treated with a sulfuric acid (H_2_SO_4_) solution with a 3:1 ration for 2 hours at 60 °C. After being washed with ultrapure water, the sensors were incubated in a 2% polyethyleneimine–average molecular weight 4000 (Sigma-Aldrich, São Paulo, Brasil Ltd.) solution in dimethyl sulfoxide (DMSO) (Sigma-Aldrich, St. Louis, MO, USA) for 24 hours at room temperature. For crosslinking the amino group and fixing the antibodies, the sensors were incubated in a 2.5% glutaraldehyde (Sigma-Aldrich, USA) solution in a pH 7.0 phosphate buffer for 24 hours at 37 °C. Then, the sensors were washed in a phosphate buffer with a pH 7.0 and dried overnight at 30 °C.

The next step was the fixation of the antibody to the amine radicals immobilized earlier on the fiber surface. The sensors were incubated with protein A (Sigma-Aldrich, St. Louis, MO, USA) for one hour at 30 °C, followed by incubation for four hours in a solution of 0.1 mg/mL of anti-*E. coli* antibodies (Bio-Rad, Watford, UK).

### 3.3. Bacteria Suspension and E. coli Detection Procedures

*E. coli* O55 bacterial strains were used to prepare the suspensions that were cultivated in Tryptic Soy Agar (Merck, Darmstadt, Germany) and incubated for 24 hours at 37 °C. Subsequently, bacterial suspensions were prepared by adding colonies into a tube containing 10 mL of 0.85% saline solution. The tube was vortexed for homogenization and turbidity compared with the McFarland scale, where a reading of 0.5 is equivalent to 10^8^ CFU/mL. In the other tubes, 9 mL of an 0.85% saline solution were added to prepare suspensions diluted to 10^4^ CFU/mL by adding 1 mL of the previously diluted solution.

For bacterial detection, we prepared five beakers with 10^8^ CFU/mL of suspension and another five beakers with 10^4^ CFU/mL suspension to produce five response curves for the 10^8^ CFU/mL and five response curves for the 10^4^ CFU/mL.

### 3.4. The Optoelectronic System

Figure 2 shows the block diagram of the developed electronic instrumentation. The sensing head employed for housing the reference sensor and immunosensor was made with aluminum. The head accommodates both the input and output ends of U-Shaped POF s, one functionalized with antibodies for bacterial detection, and another one not functionalized, to be used as a reference sensor. The use of a single LED as the light source for both U-Shaped POF sensors compensates for eventual optical power fluctuations.

The output signal from the two photodetectors employed were connected to low-noise FET-input op-amps arranged as transimpedance amplifiers (TL072, Texas Instruments Inc., Dallas, TX, USA) providing two voltage outputs, one as the reference, and the other as the sensing signal itself. Each signal was connected to the inputs of an instrumentation amplifier (INA121, Texas Instruments Inc.) that performs a differential measurement. The output signal was then filtered and sampled by a 16 bit resolution analog-to-digital converter (DAQ USB-6002, National Instruments Inc., Austin, TX, USA). From a 16 bit digital-to-analog converter, a current source was implemented to control the LED.

A dedicated Graphical User Interface (GUI) was built in LabVIEW 2018 Student Edition to support the optoelectronic instrumentation. Figure 3 shows the front panel view of the software.

### 3.5. Measurement Methodology

When bacteria in the water adhere to the fiber surface, the RI of the surrounding media changes from between 1.33 (pure water) up to a maximum of 1.39 (pure bacteria). Measuring the effect of this small RI variation on the sensor needed a high gain with a very high signal-to-noise ratio, see [6]. To circumvent the small difference in the output voltage of the electronic system, a new methodology was tested to generate larger differences between sensors with bacteria and the sensors without bacteria. If the sensors were read in contact with air, the presence of bacteria, with a refractive index (RI) of 1.39, contrasted better with air with an RI of 1 when compared to making measurements in water with an RI = 1.33. Therefore, the sensing methodology adopted, shown in Figure 4, is as follows: (A) Calibration of the reference sensor and the immunosensor in air (RI = 1). This is done by adjusting the gains of the sensor amplifier and the reference amplifier to produce the same output. (B) Both sensors are immersed in the *E. coli* suspension for 10 min. (C) Sensors are suspended from the beaker to the air and read again by the system.

### 3.6. Test and Simulation of the Methodology

To verify that the sensor method provides real results, a confirmation test was performed with the following methodology: Two U-Shape sensors, an uncladded POF as a reference and a conventional pristine cladded POF simulating the presence of bacteria around the fiber, were inserted, and removed from a beaker containing pure water. The results of this test are shown in Section 4.

## 4. Results

### 4.1. Results of the Immunocapture

After all the functionalization processes and RI measurements, four sensor pairs were used to detect *E. coli* at 10^8^ CFU/mL and another four sensor pairs at 10^4^ CFU/mL. Figure 5 and Figure 6 show the results of the four sensors with 10^8^ CFU/mL and the four sensors with 10^4^ CFU/mL. Notice that, in both cases, the reference sensor output returns to the same level it was before immersion in the analyte, as expected, whereas the immunosensor shows a higher level after immersion due to the bacteria adhered to its surface.

The results for all sensors were similar, with the immunosensor presenting a higher output after detection; however, due to the larger number of bacteria adhered to the sensor in the 10^8^ CFU/mL analyte, the output power was higher because more light was transmitted by the fiber than that of the 10^4^ CFU/mL analyte. This behavior at both bacterial concentrations was expected because the surrounding bacteria around the fiber acts as a fiber core, allowing the fiber to transmit more propagation modes.

In this research, we were interested in ensuring detection capability. A bacterial concentration of 10^8^ CFU/mL is higher than occurs in nature. This concentration is normally used to test a sensor as if the sensor does not respond to this concentration, it will not work at the *E. coli* concentration normally found in contaminated water, which is between 10^1^ and 10^2^ CFU/mL [28]. After validating the sensor’s ability to detect *E. coli*, the next step is to lower the concentration further to find the limit of detection (LoD).

This experiment confirmed that the sensor can detect 10^4^ CFU/mL, and experiments will be continued to further reduce the bacterial concentration. The results in Figure 6 show that the response of the sensor to 10^4^ CFU/mL is still much higher than that of the reference, indicating that there is still scope for detecting lower concentrations. Table 1 shows the average and uncertainty of the results of the four sensors after detecting concentrations of 10^4^ and 10^8^ CFU/mL. Note that the two measurements do not overlap, indicating that there is still room for detecting an intermediate concentration or for detecting concentrations lower than 10^4^ CFU/mL, which is the goal of new ongoing experiments.

### 4.2. Results of the Simulation of Immunosensor Behavior

In Figure 5 and Figure 6, one can see that after removing the two sensors from the water sample, the reference sensor returns to the air under exactly the same conditions as before the first immersion, which can be clearly seen from the results, because the output power is the same as before the immersion. However, the functionalized sensor returns to the air with a layer of bacteria around it, which is due to the immunocapture. So, the question arises, why does the output power of this sensor increase more than that of the reference sensor?

To better understand the behavior of the U-Shape sensor when bacteria adhere to its surface, we conducted the following experiment.

Two U-Shape sensors, an uncladded POF as a reference and a conventional pristine cladded POF simulating the immunosensor after bacterial adhesion, were tested in the system. The cladding of this sensor simulates the adhering bacteria.

The PMMA fiber core has an RI of 1.49 and the fluorinated polymer cladding has an RI of approximately 1.40.

In this experiment, the two U-Shape sensors were illuminated with the same LED in the circuit shown in Figure 2 and placed in a beaker of pure water, then exposed to air and then immersed in water again.

Due to the fiber curvature at a diameter of 10 mm, the two sensors lose light both in water and in air. When immersed in water, the two sensors produced approximately the same output power, as the outer RI of the uncladded fiber was 1.33 (water), while the outer RI of the pristine fiber was 1.40 (the RI of the cladding), as shown in Figure 7.

When both sensors were brought out of the water into the air, the outer RI of the uncladded sensor changed to 1 (RI of the air), while the outer RI of the cladded sensor remained unchanged (1.40, the RI of the cladding). However, the cladded fiber increased its optical conductivity more than that of the uncladded sensor because the air outside the cladding forms another layer that traps the light even better on the fiber. Consequently, the cladded fiber had a higher output power than the uncladded sensor. With this analogy, it is possible to validate the behavior of the immunosensor after the detection of bacteria. The results of this test are shown in Figure 7.

### 4.3. Tests of Fluorescence in a Confocal Microscope

After functionalization with PEI, immobilization of the anti-*E. coli* antibodies, and immunocapture for 10 min in the bacterial suspension, the immunosensors were washed in PBS (phosphate-buffer saline), immersed in DAPI solution for 10 min, and submitted to confocal microscopy (Leica CTR 4000, Bioz, Inc. Palo Alto, CA, USA).

Figure 8 shows the micrograph and histogram of the DAPI-stained 1 mm U-shaped POF immunosensors after the immunocapture of *E. coli* in the bacterial suspension of 10^8^ UFC/mL. An area of approximately 1.6 × 10^5^ µm^2^ was analyzed with a total average volume of 50.81 gray values.

Figure 9 shows the micrograph and histogram of DAPI-stained 1 mm U-Shaped POF immunosensors after the immunocapture of *E. coli* in bacterial suspension of 10^4^ UFC/mL. An area of approximately 1.6 × 10^5^ µm^2^ was analyzed with a total average volume of 10.35 gray values.

The histograms presented show that the immunosensor captured the bacteria, confirming that the increase in the output power of the immunosensor was due to the presence of bacteria. Additionally, it is possible to notice from the histograms that the number of captured bacteria is higher for the 10^8^ UFC/mL concentration than for 10^4^ UFC/mL concentration.

The increase in the output power observed in response to *E. coli* detection in air was because the bacteria captured by the antibody act as a new fiber cladding, which guides the light inside the fiber, in contrast with the reference sensor, which is without a bacterial cladding.

Although bacterial concentrations found in real-world samples, such as in water potability assessments, can range in much smaller concentrations (typically from 10^2^ to 10^1^ CFU/mL) than those tested in this work, the results obtained show that the immunosensor and measurement methodology proposed here could be developed to detect *E. coli* in suspension at concentrations lower than 10^4^ CFU/mL. This is an important goal that is currently being assessed by our group through improvements in the proposed immunosensor’s sensitivity, as well as in the optoelectronic system to allow for the detection of bacteria at lower concentrations. One of the improvements that will be implemented, for instance, is based on different sensor shapes, as already demonstrated in a recent study [29]. It is important to note, however, that the detection of lower concentrations is also limited by the sensitivity of the antibody provided by the manufacturer.

### 4.4. Tests of Selectivity

An important feature of sensors in general is their ability to detect the parameter for which they are designed while not detecting other parameters that may be present in the environment. In the case of the detection of *E. coli* in water, it is necessary to test the selectivity of the sensor towards other bacteria that could be present in the sample in order not to be confused with a false response. In this case, we tested Sensor 1 with *Enterobacter cloacae*, *Salmonella typhimurium*, and *Bacillus subtilis*, all at a concentration of 10^8^ CFU/mL. These tests were performed first, before *E. coli* was tested. These results, shown in Figure 10, confirm the selectivity of the antibody, which is only sensitive to *E. coli*.

## 5. Discussion and Conclusions

As already mentioned, plastic optical fibers were first developed by DuPont in 1963 and used for short-distance illumination [18]. Due to transmission losses, they were soon replaced by silica fibers and the focus of POF application shifted to sensor development. POFs can be connected to readily available transmission components, resulting in low-cost devices. Due to these unique properties, POFs have been used widely in physical, chemical, and biological sensing, as well as in strain, temperature, and displacement measurements. The pioneering study by Beres et al. (2011) [19] on the detection of *E. coli* in water using a tapered POF sensor was taken to a new level in this study by not only detecting but also quantifying *E. coli*. Subsequently, Wandermur et al. (2014) [6] developed a U-Shaped POF sensor in an electronic platform for the rapid detection of bacteria. Following these studies, Rodrigues et al. (2017) [20] investigated the sensitivity of different forms of a U-Shaped POF sensor and searched for a better efficiency at a low bacterial dilution, while Lopes et al. (2018) [21] used a specific U-Shape sensor format for the detection of sulfate-reducing bacteria, such as *Desulfovibrio alaskensis.* In this paper we reinforce the use of POFs for the detection of specific bacterial species and the open the door for quantifying them.

The experimental results were reproducible, confirming that both the immunosensors and measurement method were reliable. Moreover, the histograms of the fluorescence intensity distribution of the sensor surface confirm the results obtained by the immunosensors and show that the sensor has de facto captured bacteria that caused the observed increase in the output signals. Additionally, it was confirmed that this methodology of sensor functionalization is selective and does not produce false responses to other bacteria present in the sample.

The new method of reading the sensor in air rather than in water improved the methodology. The reason for this is that outside the water, the external RI is 1 (air), which makes a greater difference to the RI of the adhering bacteria than inside the water, where the external RI of water is 1.33, which is very close to that of the bacteria (1.39).

When comparing this new sensor system to others with a similar sensitivity, response time, and detection limit, the main advantages of this approach are the simplicity of the system, the low manufacturing costs, and the size, which allow for easy transportation to the site of use.

This new POF-based immunosensor was able to detect the presence of *E. coli* at a concentration of 10^4^ CFU/mL within 20 min. This new method sets a new standard in sensitivity and is currently the fastest *E. coli* biosensor available and a significant improvement over conventional laboratory detection technology.

One of the next steps, for our group, is to improve the sensitivity of the sensor to enable the detection of bacteria at lower concentrations. One of the improvements to be realized, for example, is based on different sensor shapes [29].

## Figures and Tables

**Figure 1 polymers-16-00861-f001:**
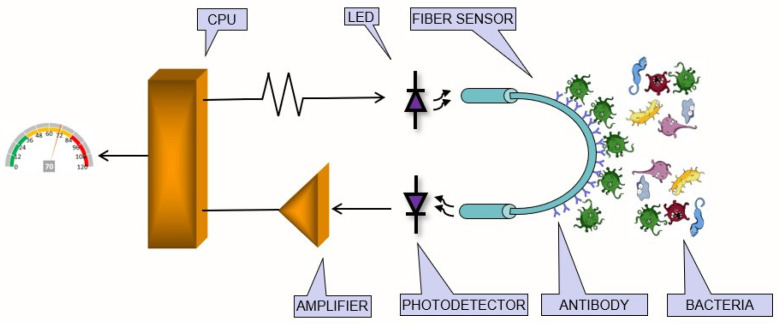
Schematic diagram of the sensing system. A CPU controls the LED output power that is coupled into the POF U-Shape. The light is guided by the fiber to the photodetector, and the output returns to the microcontroller. The fiber surface is functionalized with a specific antibody that performs the immunocapture process specifically for *E. coli*.

**Figure 2 polymers-16-00861-f002:**
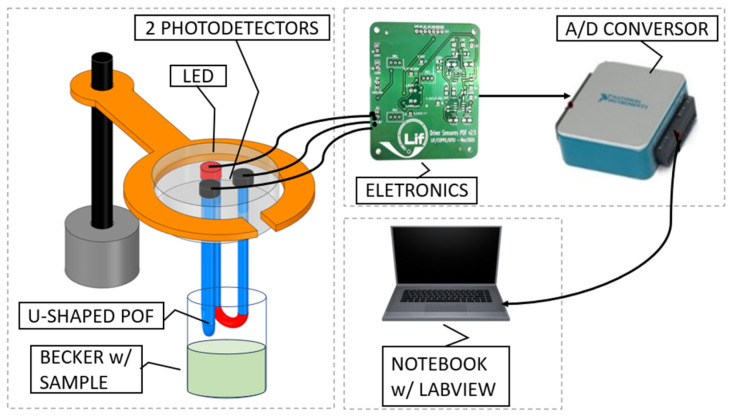
Optoelectronic setup developed for the immunosensor for sensing, signal acquisition, and conditioning.

**Figure 3 polymers-16-00861-f003:**
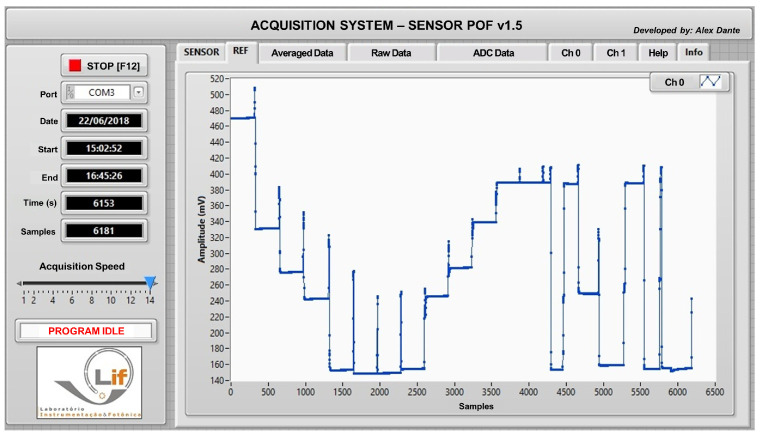
Front panel view of the software showing the response of one U-Shape sensor when immersed in water with different refractive indices (1.33, 1.35, 1.36, 1.37, 1.38, znd 1.39). The idea of this test was to check the stability and repeatability of the measurements.

**Figure 4 polymers-16-00861-f004:**
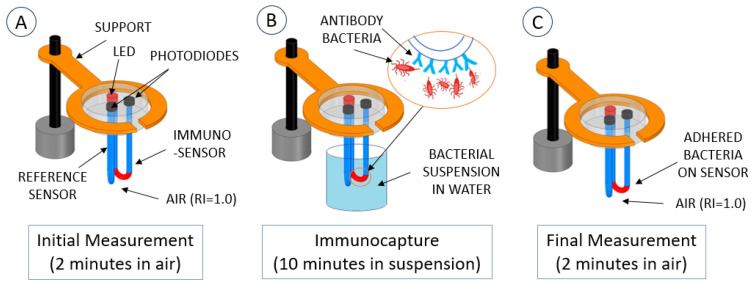
Measurement procedure for bacteria detection.

**Figure 5 polymers-16-00861-f005:**
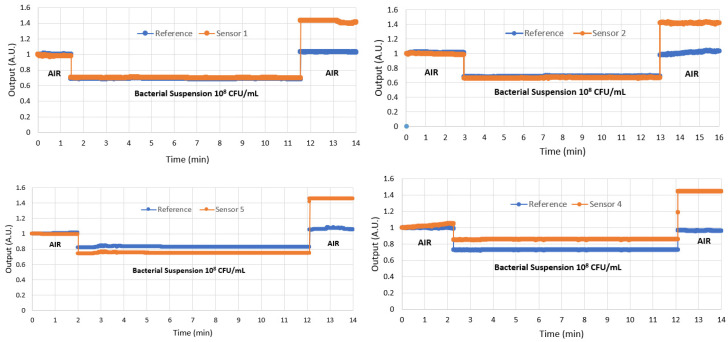
The results normalized in arbitrary units (a.u.) of the four sensors in the detection of *E. coli* in the bacterial suspension of 10^8^ CFU/mL.

**Figure 6 polymers-16-00861-f006:**
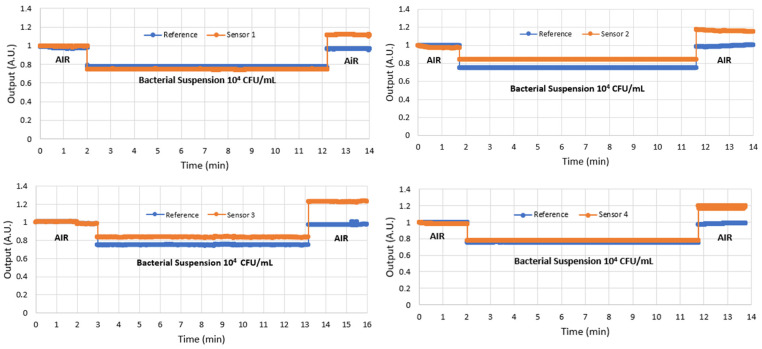
The results normalized in arbitrary units (a.u.) of the four sensors in the detection of *E. coli* in the bacterial suspension of 10^4^ CFU/mL.

**Figure 7 polymers-16-00861-f007:**
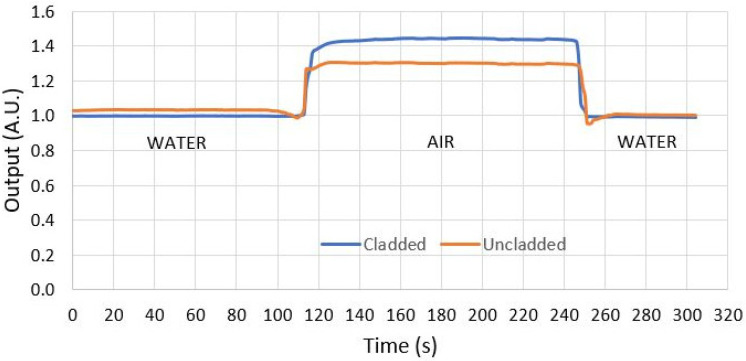
Results (in normalized arbitrary units) of the simulation test with two U-Shape sensors, an uncladded POF as a reference and a conventional pristine cladded POF simulating the immunosensor after bacteria adhesion. The cladded fiber increased its guiding capability more than the uncladded fiber due to the air outside the cladding, and therefore showed a higher output power than the uncladded fiber.

**Figure 8 polymers-16-00861-f008:**
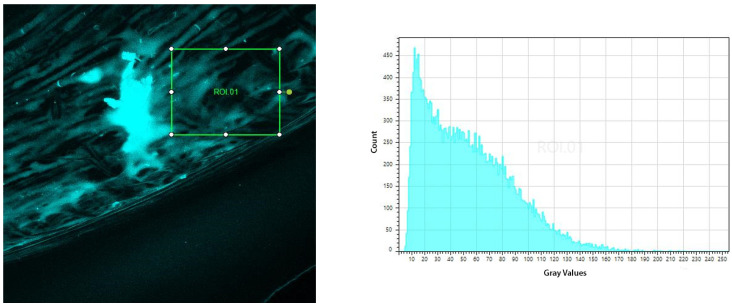
(**Left**) Micrograph of a 3D reconstruction of the DAPI-labeled POF immunosensor after 10 min in the bacterial suspension of 10^8^ CFU/mL. The inserted square represents the area of 1.6 × 10^5^ µm^2^ at 10x magnification, submitted to the evaluation of fluorescence intensity. (**Right**) Graph of the fluorescence intensity distribution of the evaluated area.

**Figure 9 polymers-16-00861-f009:**
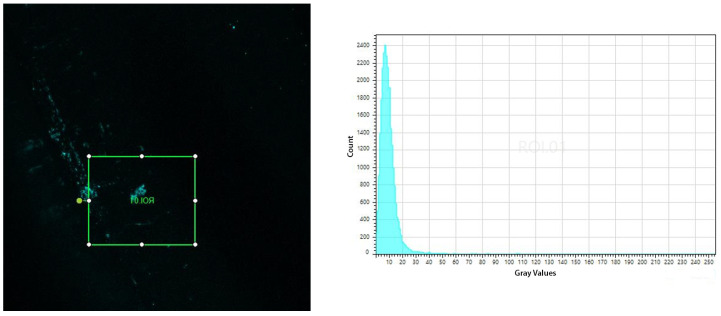
(**Left**) Micrograph of a 3D reconstruction of the DAPI-labeled POF immunosensor after 10 min in the bacterial suspension of 10^4^ CFU/mL. The inserted square represents the area of 1.6 × 10^5^ µm^2^ at 10x magnification, submitted to the evaluation of fluorescence intensity. (**Right**) Graph of the fluorescence intensity distribution of the evaluated area.

**Figure 10 polymers-16-00861-f010:**
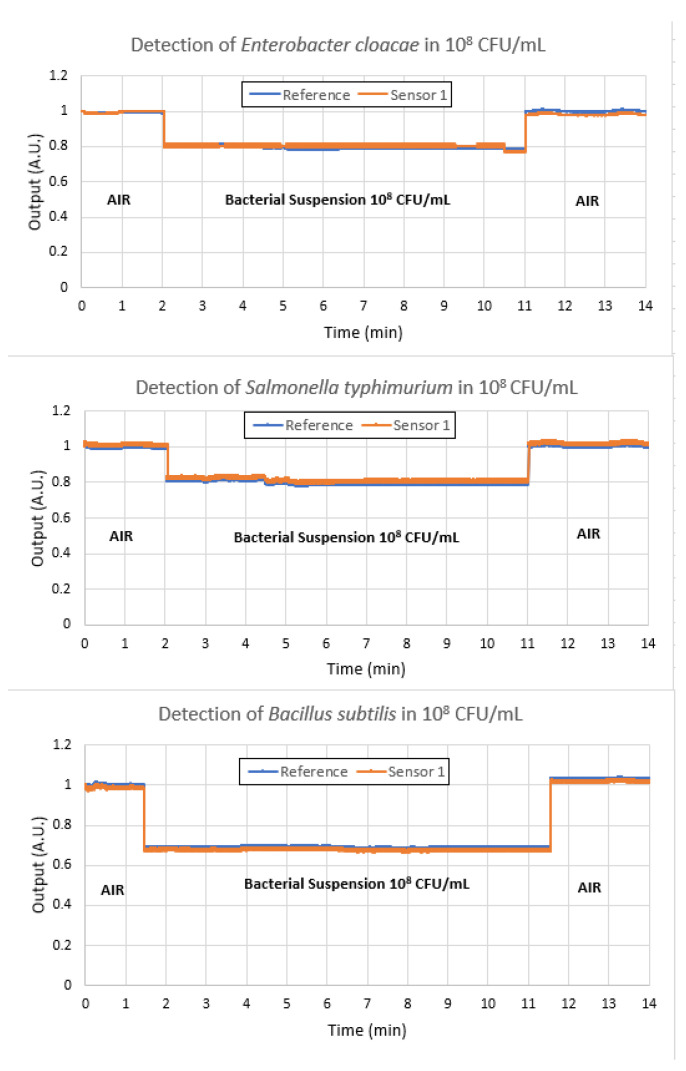
Selectivity tests of Sensor 1 with *Enterobacter cloacae*, *Salmonella typhimurium*, and *Bacillus subtilis*, all at a concentration of 10^8^ CFU/mL, showing that the antibody used is insensitive to these bacteria.

**Table 1 polymers-16-00861-t001:** Average and uncertainty of the results of the four sensors when detecting *E. coli*.

Concentration	Average of Results (a.u.) ^1^	Standard Deviation
10^4^ CFU/mL	1.12	0.05
10^8^ CFU/mL	1.33	0.05

^1^ arbitrary unit.

## Data Availability

Data are contained within the article.

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
