# Peer review of "Development of an Immunocapture-Based Polymeric Optical Fiber Sensor for Bacterial Detection in Water"

_polymers, 2024, doi:10.3390/polym16060861_

Round 1

Reviewer 1 Report

Comments and Suggestions for Authors

Dear Editor,                                                                                                                12-Jan-2024

Developing POC testing systems for pathogens, bacteria, and viruses is of great interest and suggested as the need for next-generation health management strategies. The authors developed an immunocapture-based polymeric optical fiber sensor for the detection of Escherichia coli. This optical fiber sensor sensitively detects E. coli as low as 104 CFU/mL. The selection of a sensing system is well-justified and supported by the literature. Overall, this is good research but needs a revision before recommending it for publication.

Comments:

1.      The introduction part seems too lengthy which may be boring to readers. Thus, try to shorten it.

2.      Although the sensitive 104 CFU/mL for E. coli, please provide a comparative table.

3.      This method is utilized for the detection of E. coli. However, the authors should provide the specificity test because various bacteria are present in the water. Therefore, please conduct the specificity test using different bacteria.

4.      Some minor typos are in the whole manuscript… The manuscript must be proof-read by a native English speaker or language editing service before resubmission.

Comments on the Quality of English Language

Minor editing of English language required.

Author Response

Reviewer 1

Comments and Suggestions for Authors

Dear Editor, 12-Jan-2024

Developing POC testing systems for pathogens, bacteria, and viruses is of great interest and suggested as the need for next-generation health management strategies. The authors developed an immunocapture-based polymeric optical fiber sensor for the detection of Escherichia coli. This optical fiber sensor sensitively detects E. coli as low as 10 CFU/mL. The selection of a sensing system is well-justified and supported by the literature. Overall, this is good research but needs a revision before recommending it for publication.

Comments:

  1. The introduction part seems too lengthy which may be boring to readers. Thus, try to shorten it.

Response 1: We shortened the introduction removing redundancies and repeated statements.

  1. Although the sensitive 104 CFU/mL for E. coli, please provide a comparative table.

Response 2: At the actual stage of our research, we are interested to ensuring the detection capability of the sensor. This is the reason we used that much bacteria concentration, such as 10^8 and 10^4 CFU/mL. To produce a comparative table and a graph is exactly our plan for the experiments now in progress. However as for these initial results described in this paper, we just have two points for the curve (10^4 and 10^8 CFU/mL) and therefore it is impossible to know at the moment the tendency of this curve, probably a log-like curve, but we would need more points to display it.

  1. This method is utilized for the detection of E. coli. However, the authors should provide the specificity test because various bacteria are present in the water. Therefore, please conduct the specificity test using different bacteria.

Response 3: Selectivity essays were done with Enterobacter cloacae, Salmonella typhimurium and Bacillus subtilis in 10^8 CFU/mL but was not included in the paper just to save space. It is now included in the reviewed article.

  1. Some minor typos are in the whole manuscript…The manuscript must be proof-read by a native English speaker or language editing service before re submission.

Response 4: We have proofread the manuscript removing the typos and produced a review version.

Reviewer 2 Report

Comments and Suggestions for Authors

In this study, the author developed a plastic optical fiber immunosensor using two POF U-Shape probes for the detection of bacterial. I think there are several important issues that need to be addressed before considering its publication.

1. The author did not optimize some reaction conditions when constructing the POF immunosensor, such as the potential impact of antibody loading amount on the sensor performance.

2. As an immunosensor, it did not detect bacteria of different concentrations and did not obtain a linear detection range. Additional experimental data is needed.

3. The author did not investigate the specificity, and how to determine whether the signal change of the POF immunosensors is caused by E. Coli? The author needs to supplement experiments related to sensor selectivity.

Comments on the Quality of English Language

Minor editing of English language required.

Author Response

Reviewer 2

Comments and Suggestions for Authors

In this study, the author developed a plastic optical fiber immunosensor using two POF U-Shape probes for the detection of bacterial. I think there are several important issues that need to be addressed before considering its publication.

  1. The author did not optimize some reaction conditions when constructing the POF immunosensor, such as the potential impact of antibody loading amount on the sensor performance.

Response 1: At the actual stage of our research, we are interested to assure the detection capability of the sensor. This is the reason we used that much bacteria concentration, such as 10^8 and 10^4 CFU/mL. In the next steps, we will decrease the concentration until reaching the limit of detection (LoD). Then comes the optimization of the reactions such as antibody amount.

  1. As an immunosensor, it did not detect bacteria of different concentrations and did not obtain a linear detection range. Additional experimental data is needed.

Response 2: Again, on the actual stage of our research, we are interested in ensuring the detection capability of the sensor. We did it with success for 10^8 and 10^4 CFU/mL, obtaining so far two points on the graph output voltage vs. bacteria concentration. More points are necessary to construct a calibration graph, with its trend line and LoD.

  1. The author did not investigate the specificity, and how to determine whether the signal change of the POF immune sensors is caused by Coli? The author needs to supplement experiments related to sensor selectivity.

Response 3: Selectivity essays were done with Enterobacter cloacae, Salmonella typhimurium and Bacillus subtilis in 10^8 CFU/mL but were not included in the paper just to save space. It is now included in the reviewed article.

Reviewer 3 Report

Comments and Suggestions for Authors

Test-system for E. coli assay has been developed in the article. Antibody and protein A has been immobilized on the optical detector by polyethyleneimine modified with glutaraldehyde for bacteria detection. Authors have improved detection protocols, water calibration has been replaced by air calibration. The article provides interesting results for rapid pathogen detection but some details should be improve in article.

1. For approbation of the reported test-system, E. coli concentrations were chosen as 104  and 108 CFU/mL. Could you give arguments of this concentrations choosing.

2. In the 4.1. section, repeatability results are presented for 104  and 108 CFU/mL E.coli detection. For future comparing sensor performances could you estimate relative standard deviation (%) of refractive index after 104  and 108 CFU/mL E.coli detection. For detection 108 CFU/mL E.coli this parameter would be small, because according to fig. 5, refractive index is close to 1.4. As for 104 CFU/mL E.coli detection generally refractive index is about 1.2, but one of the measurement less then 1.2 (fig 6, one of the pic with axis time has red, others are black). After estimation of relative standard deviation it could be observed significance of this fluctuations.

3. Fig 5 and 6 are included four pics. they should be marked as a,b,c and d. Put description in name of this figs.

4. Section 4.2 is confusing, because reported results (RI of 1.49; 1.40 and then 1.33 (water) and fig. 7 are not correlate. This part should be improve.

5. Results with calibration in air should be done more informative. In Abstract authors concluded that this way of detection provide better sensitive and faster detection time. Sensitive was discussed in 4.2 section insufficiently, but there is no date of analysis time improvement.

6. 2 Section should be improved with adding of principle of modification Optical Fiber by modified polyethyleneimine and chemical reason of localization of antibody and protein A should be noted. Focuses should be done on the way of modification of surfaces using polymers at first and sensing applications is the secondary.

7. Please, add to conclusion parts the final idea of such optical device. Is the prototype would be semiquantitative and detect only over limits E. coli content?

8. Please, add the confocal microscopy main results in abstract and compress introduction part in abstract.

9. In abstract you submit that detection time is less than 10 minute, but in the fig. 5, 6 only incubation time is 10 min, then it should be measure refractive index in the air (is about 2-3 mins, because of stabilization of the RI) and the sum time of measurement is a little bit more then 10 min.

10. Please, add to section 3 information about polyethyleneimine (Molecular weight, name of brand and country), source of E. coli O55 bacterial strains.

In general, the work is important in the context of the development of rapid monitoring of E.coli content and is of interest to the scientific community. Such work may be allowed to be published, but with corrections and suggested improvements.

Author Response

Reviewer 3

Comments and Suggestions for Authors

Test-system for E. coli assay has been developed in the article. Antibody and protein A has been immobilized on the optical detector by polyethyleneimine modified with glutaraldehyde forbacteria detection. Authors have improved detection protocols, water calibration has been replaced by air calibration. The article provides interesting results for rapid pathogen detection but some details should be improve in article.

  1. For approbation of the reported test-system, coli concentrations were chosen as 10 4 and 10 8 CFU/mL. Could you give arguments of this concentrations choosing.

Response 1: In the current phase of our research, we are interested in ensuring the sensor's detection capability. For this reason, we have used bacterial concentrations as high as 10^8 and 10^4 CFU/mL. A bacterial concentration of 10^8 CFU/mL is by far more than occurs in nature. This concentration is normally used to test a sensor. If the sensor does not respond to this concentration, it will not work at the E. coli concentration normally found in contaminated water, which is between 10^1 and 10^2 CFU/mL (*). After validating the sensor's ability to detect E. coli, the next step is to lower the concentration further to find the limit of detection (LoD). In this experiment, it was confirmed that the sensor is capable of detecting 10^4 CFU/mL, and experiments will continue to lower the bacterial concentration even further. The results in Fig. 6 show that the response of the sensor to 10^4 CFU/mL is still much higher than that of the reference, indicating that there is still room to detect lower concentrations.

(This text was included in the manuscript.)

(*) Odonkor ST, Mahami T. Escherichia coli as a Tool for Disease Risk Assessment of Drinking Water Sources. Int J Microbiol. vol. 2020, Article ID 2534130, 7 pages, 2020. https://doi.org/10.1155/2020/2534130

  1. In the 4.1. section, repeatability results are presented for 10 4and 10 8CFU/mL coli detection. For future comparing sensor performances could you estimate relative standard deviation (%) of refractive index after 10 4and 10 8CFU/mL E. coli detection. For detection 10 8CFU/mL E. coli this parameter would be small, because according to fig. 5, refractive index is close to 1.4. As for10 4 CFU/mL E. coli detection generally refractive index is about1.2, but one of the measurement less then 1.2 (fig 6, one of the pic with axis time has red, others are black). After estimation of relative standard deviation it could be observed significance of this fluctuations.

Response 2: Thank you for the suggestion. The following text was included in the manuscript: Table 1 shows the average and uncertainty of the results of the four sensors after detecting concentrations of 104 and 108 CFU/mL concentrations. Note that the two measurements do not overlap, indicating that there is still room for detecting an intermediate concentration or for detecting concentrations lower than 104 CFU/mL, which is the goal of new ongoing experiments.

Table 1. Average and uncertainty of the results of four sensors when detecting E. coli.

Concentration

Average of results (a.u.)1

Standard deviation

104 CFU/mL

1.12

0.05

108 CFU/mL

1.33

0.05

1 arbitrary unit.

  1. Fig 5 and 6 are included four pics. they should be marked as a,b,c and d. Put description in name of this figs.

Response 3: Since we are not dealing specifically with a particular figure, we have refrained from marking each one for the sake of simplicity.

  1. Section 4.2 is confusing, because reported results (RI of 1.49;1.40 and then 1.33 (water) and fig. 7 are not correlate. This part should be improve.

Response 4: It is really confusing, particularly because we made not clear that the output responses of the sensors (Y-axis) in Fig. 5, Fig. 6 and Fig. 7 are not refractive index, as you interpreted (and the readers would too), but normalized arbitrary units. To fix this issue we informed this on each figure caption and rearranged section 4.2 to a more detailed explanation.

  1. Results with calibration in air should be done more informative. In Abstract authors concluded that this way of detection provide better sensitive and faster detection time. Sensitive was discussed in 4.2 section insufficiently, but there is no date of analysis time improvement.

Response 5: An explanation of calibration in air was incorporated in the text.

The only available data on the time taken to detect pathogens is provided by conventional methods. They rely on time-consuming enrichment steps followed by biochemical identification strategies that require assay times of 24 hours to one week [3], while the time it takes to the proposed sensor to detect bacteria is 10 minutes.

  1. 2 Section should be improved with adding of principle of modification Optical Fiber by modified polyethyleneimine and chemical reason of localization of antibody and protein A should be noted. Focuses should be done on the way of modification of surfaces using polymers at first and sensing applications is the secondary.

Response 6: Molecular layers of polyethylenimine, antibodies, or other reagents formed around the optical fiber do not alter the behavior of light traveling within the fiber. This is because the thickness of these layers is in the order of nanometers, i.e. much smaller than the wavelength of the light (micrometers). For this reason, these layers are not taken into account when we study the behavior of evanescent fields.

  1. Please, add to conclusion parts the final idea of such optical device. Is the prototype would be semiquantitative and detect only over limits coli content?

Response 7: Included.

  1. Please, add the confocal microscopy main results in abstract and compress introduction part in abstract.

Response 8: Done

  1. In abstract you submit that detection time is less than 10minute, but in the fig. 5, 6 only incubation time is 10 min, then it should be measure refractive index in the air (is about 2-3 mins, because of stabilization of the RI) and the sum time of measurement is a little bit more then 10 min.

Response 9: Corrected to 20 minutes both in the abstract and in the conclusion.

  1. Please, add to section 3 information about polyethyleneimine (Molecular weight, name of brand and country), source of E. coli O55 bacterial strains.

Response 10: Done

A thank you note: I am also a reviewer of some journals and I know how much time it takes to prepare a thorough and complete review of an article. With your comments and suggestions, the article has certainly improved and become more informative. So, with this note, I would like to thank you for your time.

In general, the work is important in the context of the development of rapid monitoring of E. coli content and is of interest to the scientific community. Such work may be allowed to be published, but with corrections and suggested improvements.

Round 2

Reviewer 1 Report

Comments and Suggestions for Authors

Dear Editor, The authors addressed all my comments. So, this manuscript is recommended for publication in its current form.

Author Response

We thank Reviewer 1 for her/his kind contribution to the success of our paper and for this very positive review.

Reviewer 2 Report

Comments and Suggestions for Authors

The author has added the data of specificity investigation and showed good selectivity. But, I still think that a detection range experiment should be conducted, since the detection range and selectivity are the two main performance parameters for a sensor.

Comments on the Quality of English Language

Minor editing of English language required.

Author Response

Reviewer 2 - Comment 1:

“The author has added the data of specificity investigation and showed good selectivity. But, I still think that a detection range experiment should be conducted, since the detection range and selectivity are the two main performance parameters for a sensor.”

Response to Comment 1

We thank Reviewer 2 for their continued efforts to improve our paper. We note that this request is essentially the same as was requested in the first round of review:

 “As an immunosensor, it did not detect bacteria of different concentrations and did not obtain a linear detection range. Additional experimental data is needed.”

Referee 2 is 100% correct to say that a “detection range” would improve the paper. What we are saying is that we have demonstrated a new technology that works, and we have presented and tested a prototype and that is what we want to publish. Presenting a detection range would involve a significant amount of new work and experiments which would not change the innovative result we wish to publish here. It would just add a linear detection range to the findings. Publication, as is, could lead to the funding of new experiments to include a detection range.  We are at the point where Reviewer 2 is right, but we request they accept our position. If he/she does not, then it will be up to the editor to decide based on their viewpoint and Reviewer 1 and Reviewer 3. We reinforce that Reviewer 2 is not wrong about the value of a detection range; the question is whether he/she can accept the paper for publication without it. We are not able to do those extra experiments now and wish to publish as is.

Reviewer 2 - Comment 2:

“Minor editing of English language required.”

Response to Comment 2

My name is Andrew Macrae, I am one of the authors and I am British. Marcelo has asked me to write this section and comment on English. I invite the Editor or Reviewer 2 to show me the minor editing he/she desires. If it is a question of style, I can change the style if someone shows me where in the text. I proofed the paper before submission and believe the English is acceptable. Thank you.

We would like to thank the editorial team and all the referees for their prompt and professional feedback. We do hope the final decision is to publish and we look forward to hearing from you soon.

Best wishes and thanks.

Marcelo

Round 3

Reviewer 2 Report

Comments and Suggestions for Authors

I look forward to seeing the detailed performance of sensors constructed based on this method in the author's subsequent papers.